# A Novel Effector Protein SsERP1 Inhibits Plant Ethylene Signaling to Promote *Sclerotinia sclerotiorum* Infection

**DOI:** 10.3390/jof7100825

**Published:** 2021-10-01

**Authors:** Hongxia Fan, Wenwen Yang, Jiayue Nie, Wenjuan Zhang, Jian Wu, Dewei Wu, Youping Wang

**Affiliations:** 1Key Laboratory of Plant Functional Genomics of the Ministry of Education, Yangzhou University, Yangzhou 225009, China; fanhongxialab407@163.com (H.F.); yww_lab407@163.com (W.Y.); jiayuenie@163.com (J.N.); 15896412540@163.com (W.Z.); wu_jian@yzu.edu.cn (J.W.); 2Jiangsu Key Laboratory of Crop Genomics and Molecular Breeding, Yangzhou University, Yangzhou 225009, China

**Keywords:** *Brassica napus*, *Sclerotinia sclerotiorum*, effector, ethylene

## Abstract

*Sclerotinia sclerotiorum* is one of the most devastating pathogens in *Brassica napus* and causes huge economic loss worldwide. Though around one hundred putative effectors have been predicted in *Sclerotinia sclerotiorum* genome, their functions are largely unknown. In this study, we cloned and characterized a novel effector, SsERP1 (ethylene pathway repressor protein 1), in *Sclerotinia sclerotiorum*. SsERP1 is a secretory protein highly expressed at the early stages of *Sclerotinia sclerotiorum* infection. Ectopic overexpression of *SsERP1* in plant leaves promoted *Sclerotinia sclerotiorum* infection, and the knockout mutants of *SsERP1* showed reduced pathogenicity but retained normal mycelial growth and sclerotium formation, suggesting that SsERP1 specifically contributes to the pathogenesis of *Sclerotinia sclerotiorum*. Transcriptome analysis indicated that SsERP1 promotes *Sclerotinia sclerotiorum* infection by inhibiting plant ethylene signaling pathway. Moreover, we showed that knocking down *SsERP1* by in vitro synthesized double-strand RNAs was able to effectively inhibit *Sclerotinia sclerotiorum* infection, which verifies the function of SsERP1 in *Sclerotinia sclerotiorum* pathogenesis and further suggests a potential strategy for Sclerotinia disease control.

## 1. Introduction

*Brassica napus* is one of the most important oil crops and contributes a considerable proportion of the world’s edible vegetable oil [1,2]. However, the quality and yield of rapeseed are seriously threatened by Sclerotinia disease caused by *Sclerotinia sclerotiorum*, a typical necrotrophic plant pathogen with a broad host range [3]. In addition to *B. napus*, *S. sclerotiorum* infects many other agronomically important crops, such as soybean and sunflower [4]. So far, no genetic source of complete resistance to *S. sclerotiorum* has been reported within Brassica species [3,5], which hampered the creation of resistant cultivars by hybrid breeding. Therefore, deciphering the mechanisms of *S. sclerotiorum* pathogenesis is crucial for developing new effective and environment friendly methods to control Sclerotinia disease.

Effectors play important roles in plant–microbe interactions and are critical for the successful infection or transmission of many plant pathogens [6,7,8]. The action mechanisms of effectors have been extensively studied in biotrophic and host-specific necrotrophic plant pathogens [7,8,9]. These effectors adopted various strategies to overcome plant immune systems or manipulate plant-microbe-vector tritrophic interactions. For example, the *Pseudomonas syringae* effector HopA1 inactivates plant mitogen-activated protein kinases to repress pathogen-associated molecular pattern (PAMP) -induced plant immunity [10], and the *Cucumber mosaic virus*-encoded protein 2b not only suppresses plant RNA silencing machinery to facilitate infection but also attenuates plant jasmonate signaling to promote insect vector-mediated transmission [11]. However, compared with that in biotrophic and host-specific plant pathogens, the functions of effectors in broad host range necrotrophic plant pathogens are poorly understood [7,8].

Accumulating evidence suggests that, in addition to secreting toxins (for example, oxalic acid) and cell-wall-degrading enzymes, *S. sclerotiorum* and other broad host range necrotrophic plant pathogens also employ protein or RNA effectors to counteract host immune system for successful infection [4,12,13]. The recently released genome sequencing data of *S. sclerotiorum* provide unprecedented opportunities for comprehensive and in-depth investigation of its effectors [14,15]. Bioinformatic analysis predicted around one hundred putative effector-encoding genes in the genome of *S. sclerotiorum* [14,15,16]. However, apart from a few exceptions, most of these putative effectors have not been experimentally characterized [4,17].

In this study, we report a novel effector protein of *S. sclerotiorum*, designated as SsERP1 (ethylene pathway repressor protein 1). By ectopic overexpression, knockout mutant analysis, and spray-induced RNA silencing, we showed that SsERP1 significantly contributes to the pathogenicity of *S. sclerotiorum*. Transcriptome data further indicate that SsERP1 specifically inhibits plant ethylene signaling to promote *S. sclerotiorum* infection. Our work adds new insights into the mechanisms of effector-facilitated *S. sclerotiorum* pathogenesis.

## 2. Materials and Methods

### 2.1. Plant Growth Condition

Wild-type *Arabidopsis thaliana* (Col-0), *Nicotiana benthamiana*, and *Brassica napus* J9712 were grown in a greenhouse at 22–24 °C with a 16-h light/8-h dark cycle.

### 2.2. Yeast Secretion Assay

The function of the predicted signal peptide of SsERP1 was verified using the *YTK12* yeast secretion assay as described by Yu Yang et al. [18]. The predicted signal peptide (SP) sequence was constructed into pSUC2 vector and transformed into yeast strain *YTK12*. The cloning primer sequences are shown in Appendix A. Transformants were grown on YPDA (yeast extract 10 g/L, peptone 20 g/L, glucose 10 g/L, agar powder 20 g/L), CMD-W (amino acid-free nitrogen source YNB 6.7 g/L, Trp- deficient yeast extract 0.75 g/L, sucrose 20 g/L, glucose 1 g, agar powder 20 g/L), and YPRAA (yeast extract 10 g/L, peptone 20 g/L, raffinose 20 g/L, agar powder 20 g/L) medium to assess the secretion ability of SP. The *YTK12* strain transformed with pSUC2-Mg87 or pSUC2-Avr1b was used as negative control and positive control, respectively [19].

### 2.3. Subcellular Localization Analysis

To detect the subcellular localization of SsERP1, the coding sequences of full-length *SsERP1* and the signal peptide-truncated *SsERP1* (*SsERP1DSP*) were cloned into *pJG186* vector. The cloning primer sequences are shown in Appendix A. The resulting constructs (*pJG186*-*SsERP1* and *pJG186*-*SsERP1DSP*) were transformed into *Agrobacterium* strain GV3101, and transiently expressed in *N. benthamiana* leaves by *Agrobacterium* infiltration. The subcellular localization of GFP-fused SsERP1 and SsERP1 were observed using a confocal microscope (LSM 880NLO) 4 days after infiltration.

### 2.4. The Effect of Ectopic SsERP1 Overexpression on S. sclerotiorum Pathogenicity

For transient expression of *SsERP1*, we used the *Tobacco rattle virus* (TRV)-based vectors *pTRV1 and pTRV2* [20]. *pTRV* vectors were previously shown to be effective in gene overexpression, with an even higher overexpression level than 35S promoter-based vectors [21]. *SsERP1* and *SsERP1DSP* were constructed into *pTRV2* vector and then transformed into *Agrobacterium* strain GV3101. The cloning primer sequences are shown in Appendix A. MES buffer (10 mM MgCl_2_, 10 mM MES, 200 μM acetosyringone, pH 5.7)-resuspended cultures of *Agrobacterium* strains harboring indicated *pTRV2* constructs were mixed with that containing *pTRV1* construct at a 1:1 ratio to make a final OD_600_ of 0.8. After incubation at room temperature in darkness for 2 h, *Agrobacterium* were injected into the leaves of *N. benthamiana*. The leaves were cut off and inoculated with *S. sclerotiorum* 48 h after *Agrobacterium* injection.

### 2.5. S. sclerotiorum Inoculation

The wild-type *S. sclerotiorum* strain used in this study is a field isolate and was maintained on PDA medium at 22 °C without light. For *S. sclerotiorum* inoculation with mycelium suspension, 6 fresh mycelium agar plugs with a diameter of 5 mm were inoculated into a 250 mL conical flask containing 150 mL PDB and incubated at 22 °C for 24 h at 150 rpm. The yielded mycelia balls were collected by filtering the overnight culture with sterilized gauzes and washed three times with ddH_2_O and PDB, respectively. The mycelia balls were smashed with, T18 digital (IAK) at 10,000 rpm for 15 min on ice. The resulting liquid mycelium suspension was adjusted to an OD_600_ of 2.0 using PDB, and 5 µL of the suspension was used for every single inoculation on plant leaves. For *S. sclerotiorum* inoculation with mycelium agar plugs, fresh mycelium agar plugs with a diameter of 5 mm were punched out from the plates and carefully inoculated onto plant leaves.

### 2.6. Transformation of S. sclerotiorum

The 796 bp upstream and 784 bp downstream flanking sequence of *SsERP1* were inserted into the two multiple cloning sites at the left and right side of the hygromycin expression cassette in *pLOB7*, respectively [22]. The cloning primer sequences are shown in Appendix A. The resulting *pLOB7*-*SsERP1* construct was transformed into wild-type *S. sclerotiorum* by PEG-mediated protoplast transformation as described previously [21]. The transformants were selected on PDA medium containing 150 mg/L hygromycin three times, and homologous *Sserp1* mutants were obtained by single ascospore isolation, which was further confirmed by PCR identification.

### 2.7. Transcriptome Analysis

Total RNAs were extracted from 0.5 g *S. sclerotiorum*-inoculated *B. napus* leaves. The quality of the RNAs was checked by electrophoresis (Appendix A). For each sample, 1 μg was taken for sequencing library preparation using the NEBNext^®^Ultra^TM^ RNA Library Prep Kit for Illumina^®^ (NEB, Herts, UK). The clustering of the index-coded samples was performed on a cBot Cluster Generation System using TruSeq PE Cluster Kit v4-cBot-HS (Illumia, San Diego, CA, USA). After cluster generation, the prepared libraries were sequenced on an Illumina Hiseq x-ten platform at Biomarker Technologies (Beijing, China) and paired-end reads were generated. Differential expression analysis was performed using the DESeq R package (1.10.1). The resulting *p* values were adjusted using Benjamini and Hochberg’s approach to controlling the false discovery rate (FDR). The differently expressed genes (DEGs) were filtered out by setting the cut-off of FDR and fold-change at 0.01 and 2, respectively. KOBAS was used to perform KEGG enrichment of the differential expression genes.

### 2.8. Double-Stranded RNAs (dsRNAs) -Mediated Gene Knockdown

The target sequences were amplified (with T7 promoter) from *S. sclerotiorum* cDNA using primers shown in Appendix A (primer SsERP1A1 and primer SsERP1B1 for SsERP1A1, and primer SsERP1A2 and primer SsERP1B2 for SsERP1A2) to serve as the DNA templates for in vitro transcription. The dsRNAs were in vitro transcribed using T7 RNAi Transcription Kit (TR102-01, Vazyme Biotech, Nanjing, China) following the manufacturer’s instructions. For *S. sclerotiorum* inoculation, 5 μL dsRNA (900 ng/μL) was mixed with 1 mL liquid mycelium suspension (OD_600_ = 2.0), and then 10 μL liquid was pipetted onto plant leaves for each inoculation site.

### 2.9. Quantitative PCR (qPCR) Analysis

Total RNA was extracted from *S. sclerotiorum* or *S. sclerotiorum*-inoculated *N. benthamiana* leaves using Fungal Total RNA Isolation Kit (B518529, Sangon Biotech, Shanghai, China). DNase treatment and first-strand cDNA synthesis were conducted using HiScript 3 RT SuperMix for qPCR (+gDNA wiper) (R323-01, Vazyme Biotech, Nanjing, China). Quantitative real-time RCR was performed on Thermo Fisher Step One Plus instrument using Thermo Fisher PowerUp SYBR Green Master Mix (100031508, Thermo Fisher, Waltham, MA, USA). The following PCR program was used: 94 °C for 2 min, 40 cycles of 94 °C for 15 s, and 58 °C for 1 min. The internal reference genes for *S. sclerotiorum*, *B.napus,* and *N. benthamiana* were *Tubulin*, *BnActin7*, and *L25*, respectively. The relative gene expression levels were analyzed using the 2^−ΔΔCT^ method [23]. The qPCR primer sequences are shown in Appendix A. The specificity of the primers was checked by melting curve analysis (Appendix A).

### 2.10. The Statistical Analyses

The statistical analysis for lesion area, gene expression, mycelial growth, and acid production ability were performed using IBM SPSS 13.0 software. Statistical significance was tested using one-way ANOVA followed by Duncan’s test, with *p* < 0.05 being the threshold for statistical significance [24].

## 3. Results

### 3.1. SsERP1 Is a Secretory Protein Highly Expressed in the Early Stage of S. sclerotiorum Infection

To identify the putative effectors of *S. sclerotiorum,* we carried out a bioinformatical screening using the following criteria: (1) contain secretory signal peptide, (2) have increased expression during *S. sclerotiorum* infection, and (3) do not harbor cell wall degrading enzyme-like domains, according to the data available in public databases or literatures [14,25]. As a result, we identified around one hundred effector candidates, including SS1G_11468 (hereinafter SsERP1) described in this study. Consistent with the published transcriptome data, our quantitative PCR results showed that the expression of *SsERP1* was significantly induced at 3–12 h post *S. sclerotiorum* inoculation and decreased thereafter (Figure 1A), indicating that SsERP1 probably functions in the early stage of *S. sclerotiorum* infection.

Signal peptide analysis using SignalP revealed that SsERP1 contains a secretory signal peptide (SP) at its N terminal region (Figure 1B). In line with this, TMHMM 2.0 analysis found no transmembrane domain in SsERP1 (Appendix A). To verify the secretion property of SsERP1, we performed the yeast secretion assay. The predicted signal peptide of SsERP1 was constructed into the pSUC2 vector to fuse with the secretory peptide-truncated invertase gene *SUC2*. We also constructed the secretory signal of *P. sojae* Avr1b and the N-terminal of *M. oryzae* Mg87 into pSUC2 to serve as the positive and negative control, respectively. These plasmids were transformed into the yeast *YTK12* strain, which is unable to grow on the YPRAA medium with raffinose as the sole carbon source because of lacking endogenous invertase. As shown in Figure 1C, the *YTK12* strain transformed with the signal peptide sequences of SsERP1 and Avr1b, but not that transformed with the N-terminal of Mg87, regained the ability to grow on YPRAA medium, suggesting that the signal peptide sequences of SsERP1 (like that of Avr1b) was able to drive the secretion of SUC2 invertase, which degraded raffinose in YPRAA medium into simple sugars to support the growth of *YTK12*. This result indicates that the predicted signal peptide of SsERP1 is functional and SsERP1 is a secretory protein.

We next examined the subcellular localization of SsERP1 protein in plant cells. We fused the full-length (SsERP1) or signal peptide-truncated SsERP1 (SsERP1DSP) with GFP and transiently expressed them in tobacco leaves. Confocal microscopy showed that SsERP1, as well as SsERP1DSP, was distributed in both the nucleus and cytoplasm of plant cells (Figure 2).

### 3.2. Overexpression of SsERP1 Promotes S. sclerotiorum Infection

Next, we tested the effect of SsERP1 on the pathogenicity of *S. sclerotiorum*. We constructed SsERP1 and SsERP1DSP into the *pTRV2* vector, a virus backbone-based vector that was reported to be more effective than 35S promoter-based vectors in overexpressing effector genes [21]. We then transiently overexpressed *SsERP1* in tobacco leaves by agroinfiltration and inoculated *S. sclerotiorum* 48 h after the infiltration. *S. sclerotiorum* caused obviously larger lesions on tobacco leaves expressing either *SsERP1* or *SsERP1DSP* than on leaves expressing *pTRV2* empty vector (control) (Figure 3A,B). Consistently, the expression of *SsERP1* in tobacco leaves infiltrated with agrobacteria containing the *pTRV2-SsERP1* or *pTRV2-SsERP1DSP* vectors was significantly higher than that in the leaves infiltrated with the *pTRV2* empty vector 12 h after *S. sclerotiorum* inoculation (Figure 3C), which confirms that the enhanced pathogenicity of *S. sclerotiorum* resulted from the ectopic overexpression of SsERP1.

### 3.3. Knockout of SsERP1 Reduces the Pathogenicity of S. sclerotiorum

To verify the contribution of SsERP1 to the pathogenicity of *S. sclerotiorum*, we constructed *SsERP1* knockout mutants by homologous recombination using PEG-mediated protoplast transformation (Appendix A). Two homozygous *SsERP1* knockout strains (*Sserp1-1* and *Sserp1-2*) were obtained by single ascospore isolation. The complete deletion of *SsERP1* gene in *Sserp1-1* and *Sserp1-2* were confirmed by PCR (Appendix A).

First, we examined whether knocking out *SsERP1* affects the growth and development of *S. sclerotiorum*. We inoculated the *Sserp1* mutants and wild-type strain (WT) onto potato dextrose agar (PDA) and PBP (PDA supplemented with bromophenol blue, a pH indicator that will turn yellow when the pH becomes acidic). The mycelial growth rate of *Sserp1* mutants on the PDA medium was not significantly different from that of WT (Figure 4A,C), indicating that SsERP1 is not required for the in vitro growth of *S. sclerotiorum*. When grown on PBP medium, *Sserp1* mutants and WT generated a similar yellow color on their growth areas (Figure 4B,C), suggesting that the acid-producing ability of *S. sclerotiorum* is not obviously influenced by knocking out *SsERP1* (Figure 4B,C). Since sclerotium plays an important role in the life cycle of *S. sclerotiorum*, we further examined the effect of knocking out SsERP1 on the sclerotium formation of *S. sclerotiorum*. As shown in Appendix A, the morphology and number of sclerotia formed by *Sserp1-1* did not significantly differ from that of WT.

In contrast, we found that the lesion areas caused by the *Sserp1* mutants were significantly smaller than those caused by WT strain when inoculated on tobacco or rapeseed leaves (Figure 4D,E). Together with the ectopic overexpression assays shown in Figure 3, these results suggest that SsERP1 specifically contributes to the pathogenicity of *S. sclerotiorum*, but not to its in vitro mycelial growth, acid production, and sclerotium formation.

### 3.4. SsERP1 Suppresses Plant Ethylene Signaling

To gain insights into the molecular mechanisms of how SsERP1 promotes *S. sclerotiorum* infection, we carried out transcriptome analysis for *B. napus* plants infected with either *Sserp1* mutant or the wild-type strains (Appendix A). As shown in Figure 5A, 522 genes were up-regulated and 386 were down-regulated in *Sserp1* mutant-infected *B. napus* leaves compared with that infected with the wild-type strain. Subsequent KEGG pathway enrichment of these differentially expressed genes (DEGs) revealed the most significantly enriched pathway for plant hormone signaling transduction, in which 30 DEGs were included (Figure 5B,C). Strikingly, more than half (18) of these 30 DEGs are related to the ethylene signaling pathway, which is known to positively regulate plant defense against necrotrophic pathogens, including *S. sclerotiorum* (Figure 5C) [26,27,28]. In contrast to the ethylene signaling pathway, the expression of the ethylene biosynthetic genes (such as *ACS* and *ACO*) was not significantly different in *Sserp1-* and the WT strain-infected plants (Appendix A). These results suggest that SsERP1 inhibits ethylene signaling transduction.

To verify the transcriptome data, we performed qPCR analysis. Consistent with our transcriptome data, all the ethylene responsive genes detected by qPCR showed higher expression levels in plants inoculated with *Sserp1* mutants than those inoculated with the WT strain (Figure 5D), whereas no obvious trend of change was found for genes of jasmonate pathway, another phytohormone pathway required for full plant immunity against necrotrophic pathogens [29,30,31,32] (Appendix A). Moreover, we found that ectopic overexpression of *SsERP1* also significantly attenuated the expression of ethylene responsive genes (*NbCTR1*, *NbEBF2,* and *NbERS**1*) (Figure 5E), but showed significant impact on jasmonate responsive genes (*Nb**AOS, Nb**OPR3,* and *Nb**PDF1.2*) (Appendix A). These results confirm that SsERP1 specifically inhibits plant ethylene signaling.

Given the important role of the ethylene pathway in plant defense against necrotrophic pathogens [26,28], our results (Figure 3, Figure 4 and Figure 5) suggest that SsERP1 promotes *S. sclerotiorum* infection by repressing plant ethylene signaling.

### 3.5. Silencing SsERP1 by Synthesized dsRNAs Inhibits S. sclerotiorum Infection

Previous studies have shown that knocking down virulence genes by in vitro synthesized double strand RNAs (dsRNA) can inhibit pathogen infection [33]. Thus, we tested whether silencing *SsERP1* by dsRNA could attenuate *S. sclerotiorum* infection. We designed two dsRNAs (SsERP1A1 and SsERP1A2) targeting *SsERP1* (Figure 6A). Target prediction using siFi21 software showed that SsERP1A1 and SsERP1A2 were able to specifically bind to *SsERP1* without predictable off-target sites in either *S. sclerotiorum* or tobacco (Appendix A).

We synthesized the dsRNAs by in vitro transcription and then inoculated tobacco leaves with *S. sclerotiorum* pre-mixed with either SsERP1A1 or SsERP1A2 or pre-mixed with GFP-dsRNA (dsRNA targeting *GFP*) or an equal volume of water as controls. Quantitative PCR showed that the expression of *SsERP1* in tobacco leaves inoculated with either SsERP1A1- or SsERP1A2-treated *S. sclerotiorum* was significantly lower than that in the control leaves (Figure 6D), demonstrating that *SsERP1* was effectively silenced by the dsRNAs. Consistently, the lesions caused by *S. sclerotiorum* on tobacco leaves co-inoculated with SsERP1A1 or SsERP1A2 were smaller than those on the leaves co-inoculated with GFP-dsRNA or water (Figure 6B,C), suggesting that knocking down of *SsERP1* by in vitro synthesized dsRNAs can inhibit *S. sclerotiorum* infection. These results provide additional support to our conclusion that *SsERP1* contributes to *S. sclerotiorum* pathogenicity, and further suggest a potential strategy for controlling *S. sclerotiorum* disease by using in vitro synthesized dsRNAs that target *SsERP1*.

## 4. Discussion

Sclerotinia stem rot is one of the major diseases in *B. napus*. Considering the economic and environmental costs associated with fungicides, developing genetically resistant varieties is a more ideal way to cope with Sclerotinia disease. However, previous quantitative trait loci (QTL) mapping and genome-wide association studies (GWAS) indicate that the resistance of *B. napus* to *S.sclerotiorum* is a combined effect of many QTLs with small contributions [34,35,36,37], which obstructs the integration and application of these QTLs. The lack of highly resistant accessions within Brassica species also makes it challenging to identify novel genes that have major contribution to the trait. Given this predicament, it is critical to deepening our understanding of *S. sclerotiorum* pathogenesis, which would possibly provide a new prospective and inspire novel strategies for Sclerotinia disease management. Effectors are key weapons for diverse pathogens to overcoming plant immunity [6,11,38]. Compared with the extensive studies on effectors in specialized host-pathogen interactions, our understanding of effector-mediated interactions between plant and broad host range necrotrophic pathogens is limited [7,8]. Recent studies suggest that some broad host range necrotrophic fungi might also have a short biotrophic phase at the early infection stage, during which period they secrete effectors into plant cells to overcome host immunity and achieve successful colonization [14,15,39]. However, the functions of effectors encoded by *S. sclerotiorum* are largely unknown. In this study, we cloned and characterized a new *S. sclerotiorum* effector, SsERP1. By a series of biochemical and genetic assays, we demonstrated that SsERP1 is required for the full virulence of *S. sclerotiorum.*

Ethylene plays an important role in regulating plant defense against pathogens, including *S. sclerotiorum* [26,28]. A large number of ethylene pathway genes, including those involved in either ethylene biosynthesis or ethylene signaling transduction, are induced in plants upon *S. sclerotiorum* infection (Figure 5) [40,41]. Interestingly, our transcriptome data suggested that SsERP1 mainly repressed the induction of plant ethylene signaling transduction genes, but not the ethylene biosynthetic genes, during *S. sclerotiorum* infection, indicating that SsERP1 might target to some component(s) of ethylene signaling transduction processes to inhibit ethylene responses and facilitate *S. sclerotiorum* infection. However, we cannot exclude the possibility that SsERP1 inhibits ethylene biosynthetic genes at the post-transcriptional levels [42,43]. Identification of the direct target(s) of SsERP1 in plant cells is required to clearly elucidate how SsERP1 suppresses plant ethylene pathway.

Jasmonate pathway is also known to be required for full plant immunity against necrotrophic pathogens [44]. Previous studies showed that ethylene and jasmonate pathways synergistically regulate plant defense against necrotrophic pathogens, while antagonistically regulate plant development and defense against insects [29,30,31,32]. In our transcriptome data, we noted that three JA signaling pathway genes (*TIFY7*, *COI1*, and *TIFY10B*) were down-regulated in *B**. napus* leaves inoculated with the *Sserp1* strain compared to those inoculated with WT strain (Figure 5D and Appendix A). However, the expression of the vast majority of known jasmonate pathway genes was not significantly changed (Appendix A and Appendix A). For example, among the 42 *JAZ* genes expressed in oilseed rape, only *TIFY7* (*JAZ2*, BnC04g0635460) and *TIFY10B* (*JAZ9*, BnC06g0759780) were differently expressed in *Sserp1-* and WT strain-infected plants (Appendix A). In addition to *JAZ* genes, consistent with the transcriptome data, our qPCR analysis showed that the expression of other jasmonate pathway marker genes, such as *MYC2*, *AOS*, *LOX3,* and *OPR3,* was also not significantly affected by SsERP1 (Appendix A and Appendix A). These results suggest that SsERP1 specifically inhibits ethylene signaling pathway, while has no obvious impact on jasmonate pathway. Identifying the direct plant target of SsERP1 would provide insights into how SsERP1 specifically represses plant ethylene signaling pathway without perturbing jasmonate pathway. Given the critical role of jasmonate in plant defense against necrotrophic pathogens, we speculate that *S. sclerotiorum* might encode other effectors that target plant jasmonate pathway to enhance infection.

Spray-induced gene silencing (SIGS), where dsRNAs or small RNAs (sRNAs) are sprayed onto plant surface to silence the target genes of pathogens, has emerged as a new and powerful strategy for crop protection [45,46]. Recently, Mittes and colleagues developed a clay nanomaterial that could encapsulate dsRNA, which improved the stability of dsRNAs in the natural environment and prolonged their silencing activity [47]. The SIGS technology provides an alternative way to cope with Sclerotinia disease. In this study, we showed that silencing *SsERP1* by the dsRNAs SsERP1A1 and SsERP1A2 effectively inhibited *S. sclerotiorum* infection (Figure 6). Our work validates the possible application of SIGS in Sclerotinia disease control and presents the effector gene *SsERP1* as the potential target.

## 5. Conclusions

In this study, we cloned and characterized a novel effector of *S. sclerotiorum*, SsERP1 (ethylene pathway repressor protein 1), a secretory protein that is highly expressed in the early stages of *S. sclerotiorum* infection. Ectopic overexpression of *SsERP1* in plant leaves promoted *S. sclerotiorum* infection, and knockout or RANi of *SsERP1* expression reduced *S. sclerotiorum* pathogenicity without obvious impact on mycelial growth and sclerotium formation suggesting that SsERP1 specifically contributes to the pathogenicity of *S. sclerotiorum*. Transcriptome analysis further showed that SsERP1 promotes *S. sclerotiorum* infection by inhibiting plant ethylene signaling. Moreover, in vitro synthesized double-stranded RNA effectively suppressed *SsERP1* expression and attenuated *S. sclerotiorum* infection, providing a potential new strategy for *S. sclerotiorum* control.

## Figures and Tables

**Figure 1 jof-07-00825-f001:**
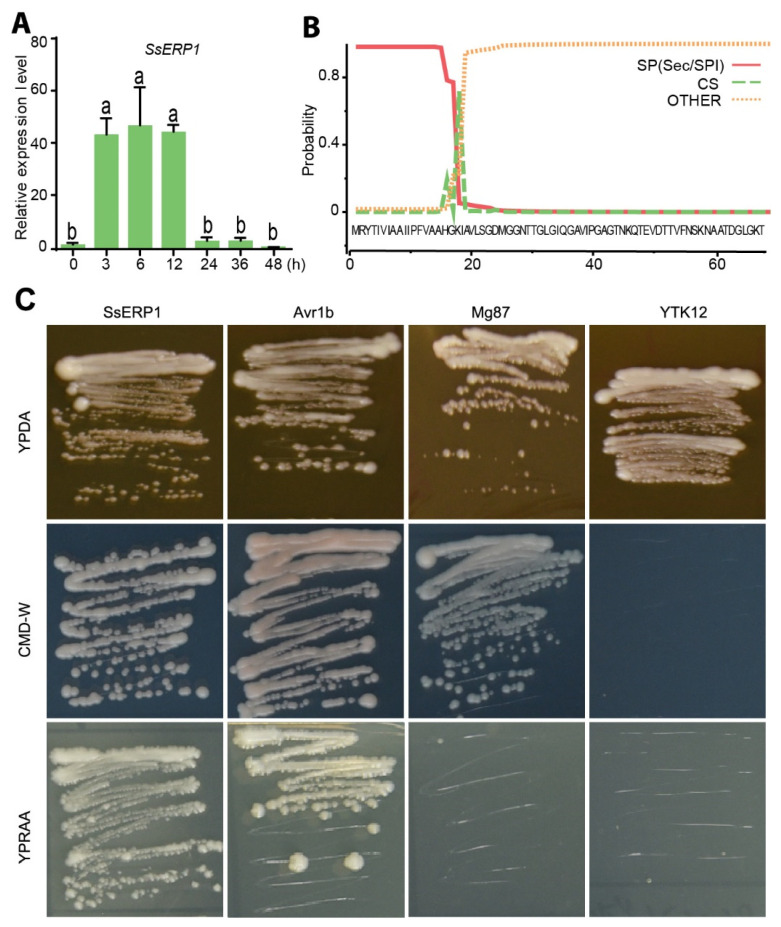
SsERP1 is a secretory protein that was highly expressed at the early stage of *Sclerotinia sclerotiorum* infection. (**A**), *SsERP1* was highly expressed at the early stage of *S. sclerotiorum* infection. The relative expression of *SsERP1* in *S. sclerotiorum*-infected tobacco leaves was detected by quantitative PCR (qPCR) at 3, 6, 12, 24, 36, and 48 h after inoculation. 0-h samples were collected immediately after inoculation. *S. sclerotiorum Tubulin* was used as an internal control. Data are shown as mean ± SD, *n* = 3. Different letters above the bars indicate statistically significant differences (*p* < 0.05, one-way ANOVA followed by Duncan’s test). (**B**), A signal peptide was predicted in SsERP1 by SignaIP-5.0. The red line denotes the probability of that the sequence contains a signal peptide. The green dotted line denotes the probability of that the sequence exists cleavage site. The red dotted line denotes the probability of that the sequence does not have a signal peptide. (**C**), The signal peptide of SsERP1 is functional in yeast secretion assay. YPDA medium was used to confirm normal growth of the yeast strains on complete medium, CMD-W medium to verify the successful transformation of indicated constructs, and YPRAA medium to assess the secretion ability of the transformed signal peptide. The *YTK12* strain was unable to grow on CMD-W and YPRAA medium, Mg87 was unable to grow on the YPRAA medium, while SsERP1 and positive control Avr1b were able to grow on all these three types of medium.

**Figure 2 jof-07-00825-f002:**
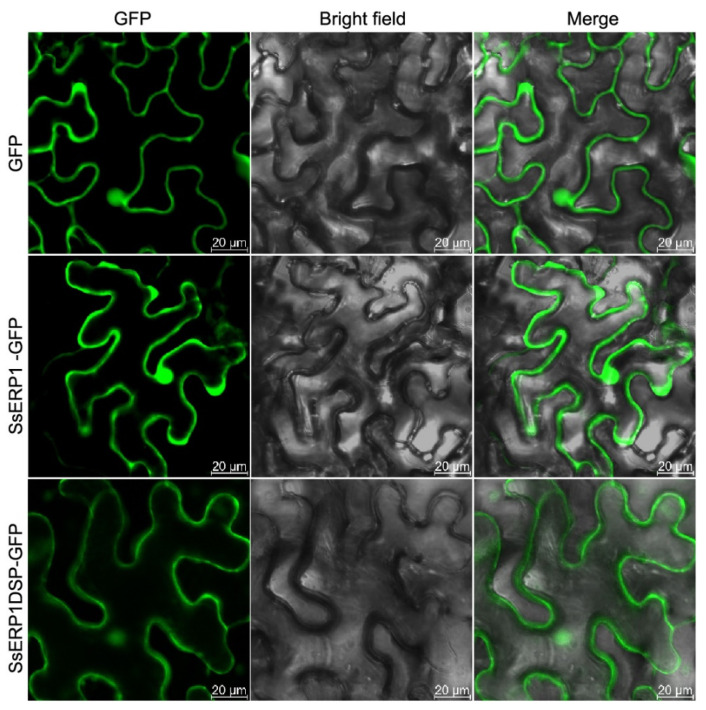
Subcellular localization of SsERP1 and SsERP1DSP. GFP-fused full-length SsERP1 and signal peptide-removed SsERP1 (SsERP1DSP) were transiently expressed in tobacco leaves by *Agrobacterium* infiltration. GFP fluorescence was observed 4 days after infiltration.

**Figure 3 jof-07-00825-f003:**
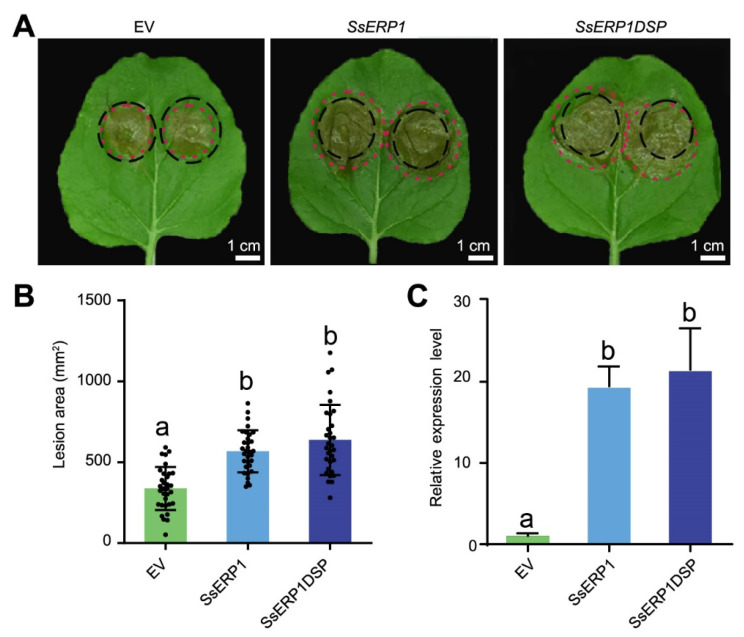
Ectopic overexpression of *SsERP1* promotes *Sclerotinia sclerotiorum* infection. (**A**), The representative phenotype of *S. sclerotiorum*-inoculated tobacco leaves. Tobacco leaves were inoculated with *S. sclerotiorum* mycelium suspension 48 h after infiltration of *Agrobacterium* containing *pTRV2-SsERP1* (*SsERP1*), *pTRV2-SsERP1DSP* (*SsERP1DSP*), or *pTRV2* empty vector (*EV*) as control. Photos were taken 48 h after *S. sclerotiorum* inoculation. Black circles indicate the leaf areas that infiltrated with indicated *Agrobacterium* and the margins of lesions caused by *S. sclerotiorum* were marked with red circles. (**B**), The lesion sizes caused by *S. sclerotiorum* in (**A**) were measured by ImageJ. Data are presented as mean ± SD, *n* = 16. Different letters above the bars indicate statistically significant differences (*p* < 0.05, one-way ANOVA followed by Duncan’s test). (**C**), qPCR analysis of *SsERP1* expression levels. Tobacco leaves were inoculated with *S. sclerotiorum* mycelium suspension 48 h after infiltration of *Agrobacterium* containing *pTRV2-SsERP1* (*SsERP1*), *pTRV2-SsERP1DSP* (*SsERP1DSP*), or *pTRV2* empty vector (*EV*) as control. Tobacco leaves were collected for qPCR analysis 12 h after *S. sclerotiorum* inoculation. Data are presented as mean ± SD, *n* = 3. Different letters above the bars indicate statistically significant differences (*p* < 0.05, one-way ANOVA followed by Duncan’s test).

**Figure 4 jof-07-00825-f004:**
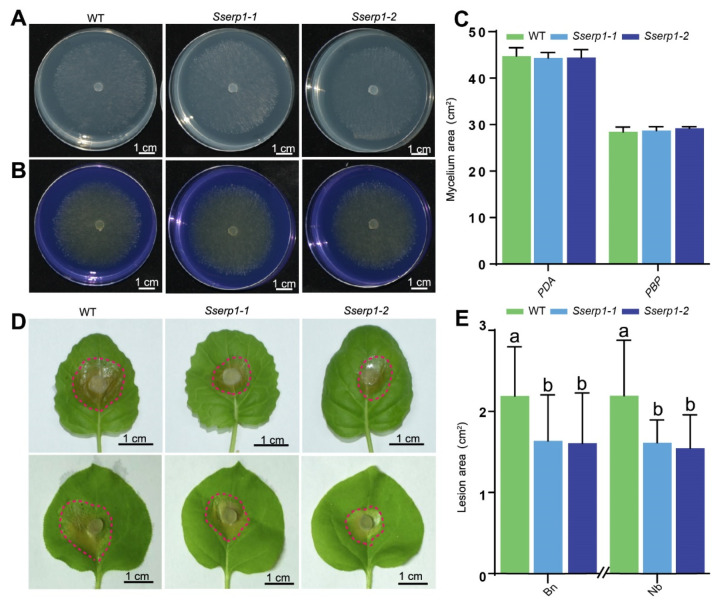
Knockout of *SsERP1* reduces the pathogenicity of *Sclerotinia sclerotiorum*. (**A**–**C**), Knockout of *SsERP1* does not affect mycelial growth (**A**,**C**) or acid production ability (**B**,**C**) of *S. sclerotiorum*. Mycelium agar plugs of the wild-type strain (WT) and *Sserp1* mutants were inoculated on PDA medium (**A**) or PBP (PDA medium containing bromophenol blue) medium (**B**) for assessing mycelial growth or acid production ability of *S. sclerotiorum,* respectively. Photos were taken 48 h after inoculation. Quantification of mycelium area on PDA and PBP plates as shown in (**C**). Data are presented as mean ± SD, *n* = 5. Different letters above the bars indicate statistically significant differences (*p* < 0.05, one-way ANOVA followed by Duncan’s test). (**D**,**E**), Knock-out mutants of *Sserp1* showed reduced pathogenicity in both *Nicotiana benthamiana* (*Nb*) and *Brassica napus* (*Bn*). Mycelium agar plugs of the wild-type strain (WT) and *Sserp1* mutants were inoculated on the left and right sides of the leaves, respectively. Photos were taken 36 h after inoculation, and the margins of lesions caused by *S. sclerotiorum* were marked with red circles (**D**). The lesion sizes were measured using ImageJ (**E**). Data are shown as mean ± SD, *n* = 9. Different letters above the bars indicate statistically significant differences (*p* < 0.05, one-way ANOVA followed by Duncan’s test).

**Figure 5 jof-07-00825-f005:**
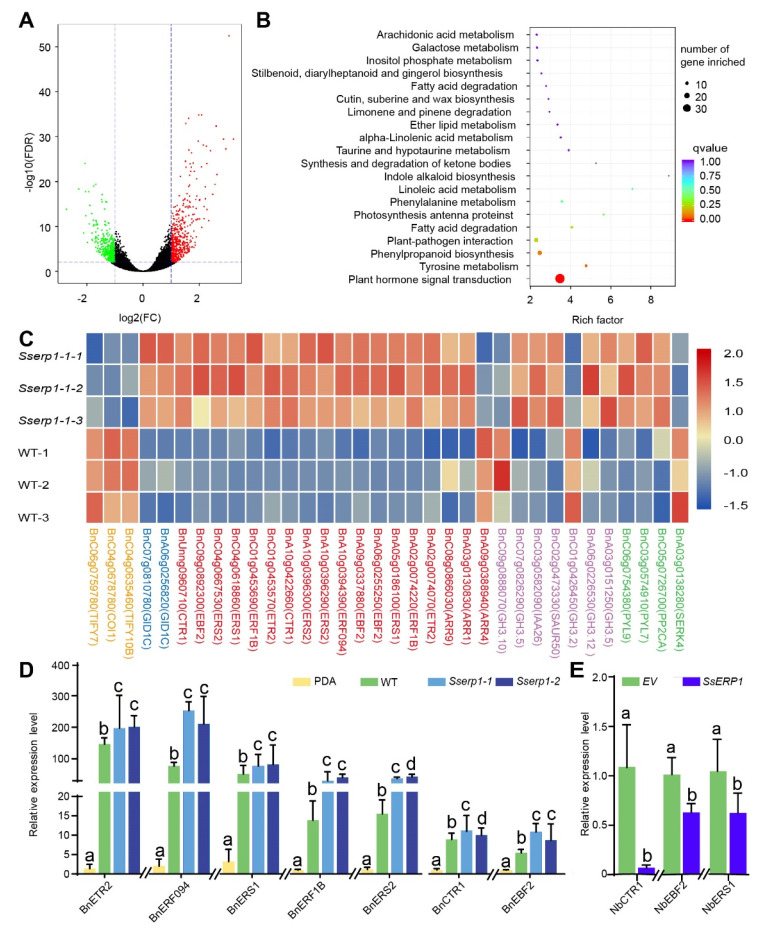
Transcriptome analysis of rapeseed leaves infected with the wild-type strain (WT) and *Sserp1* mutant strain. (**A**) Vol Plot of differentially expressed genes (DEGs) in *Sserp1* mutant and wild-type strain (WT)-infected leaves. Rapeseed leaves were inoculated with mycelium agar plugs of *Sserp1-1* or the wild-type strain, and leaves were collected for transcriptome analysis 24 h post inoculation. Up-regulated genes (*n* = 522), down-regulated genes (*n* = 386), and genes without significant change (*n* = 43,509) are shown as red, green, and black dots, respectively. (**B**), KEGG enrichment of the DEGs. The most significantly enriched pathway in this study in plants hormone signal transduction. (**C**), Heatmap of DEGs that enriched in the hormone signaling pathway. Genes involved in jasmonate, gibberellin, ethylene, auxin, or abscisic acid pathway are marked in orange, blue, red, purple, and green color, respectively. The heat map was drawn using Z-score normalized FPKM values of the DEGs by BMKCloud (www.biocloud.net, 20 August 2020). (**D**), quantitative PCR (qPCR) validation of ethylene-responsive DEGs. Rapeseed leaves were inoculated with mycelium agar plugs of *Sserp1 mutants* (*Sserp1-1* and *Sserp1-2*) or the wild-type strain (WT), or mock-treated with PDA medium (PDA), and leaves were collected for qPCR analysis 24 h post inoculation. Data are shown as mean ± SD, *n* = 3. Different letters above the bars indicate statistically significant differences (*p* < 0.05, one-way ANOVA followed by Duncan’s test). (**E**), Ectopic overexpression of *SsERP1* suppressed the expression of ethylene-responsive genes in tobacco. Tobacco leaves infiltrated with *Agrobacterium* containing *pTRV2*-*SsERP1* (*SsERP1*), or *pTRV2* empty vector (*EV*) as control, were collected for qPCR analysis 4 days after infiltration. Data are shown as mean ± SD, *n* = 3. Different letters above the bars indicate statistically significant differences (*p* < 0.05, one-way ANOVA followed by Duncan’s test).

**Figure 6 jof-07-00825-f006:**
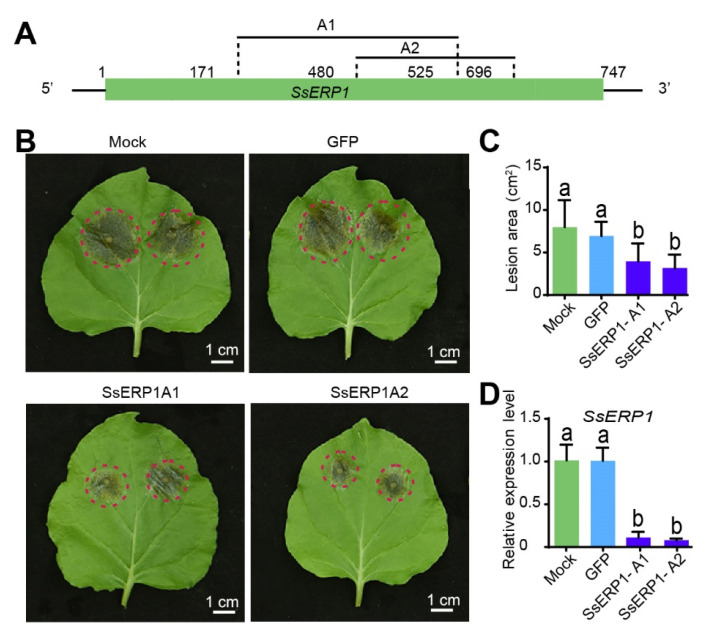
Silencing *SsERP1* by in vitro synthesized double-stranded RNAs (dsRNAs) inhibits *Sclerotinia sclerotiorum* infection. (**A**), Schematic diagram of *SsERP1* gene structure. p, promoter; sp, signal peptide, t, terminator; 5′,5′ UTR; 3′, 3′ UTR. A1 (171–525 bp) and A2 (408–696 bp) denote the target positions of the two dsRNA. (**B**–**D**), Knocking down *SsERP1* by in vitro synthesized dsRNAs attenuates *S. sclerotiorum* infection. The dsRNAs that target *SsERP1* (SsERP1A1 and SsERP1A2) or GFP (GFP), or an equal volume of RNase-free water were mixed with *S. sclerotiorum* mycelium suspensions and then inoculated on tobacco leaves. Photos were taken 2 days after inoculation. (**B**) The phenotype of inoculated tobacco leaves. The margins of lesions caused by *S. sclerotiorum* were marked with red circles. (**C**) Quantification of lesion sizes. The lesion areas were measured by ImageJ. Data are shown as mean ± SD, *n* = 20. Different letters above the bars indicate statistically significant differences (*p* < 0.05, one-way ANOVA followed by Duncan’s test). (**D**) Quantitative PCR (qPCR)analysis of *SsERP1* expression levels. Leaves were collected for qPCR analysis two days after *S. sclerotiorum* inoculation. Data are shown as mean ± SD, *n* = 3. Different letters above the bars indicate statistically significant differences (*p* < 0.05, one-way ANOVA followed by Duncan’s test).

## Data Availability

Raw Ilumina sequencing data of *Brassica nupas* are available in the NCBI Sequence Reads Archive (SRA) under the accession number PRJNA759653.

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
