# Peer review of "A Novel Effector Protein SsERP1 Inhibits Plant Ethylene Signaling to Promote Sclerotinia sclerotiorum Infection"

_jof, 2021, doi:10.3390/jof7100825_

Round 1

Reviewer 1 Report

Comments and Suggestions for Authors

The manuscript titled “A novel effector protein SsERP1 inhibits plant ethylene pathway to promote Sclerotinia sclerotiorum infection” Manuscript ID: jof-1385941, by Fan et al., provides insight on a new predicted pathogenicity effector protein secreted by the ascomycetous fungus S. sclerotiorum. In the current study, the authors cloned, characterized, and investigated the expression levels of SsERP1 (Ethylene Pathway Repressor Protein 1) of S. sclerotiorum. They used different functional genomic techniques included, including ectopic overexpression, knockout mutants, and in vitro synthesized dsRNAs, to verify the function of SsERP1 in the pathogenesis of S. sclerotiorum. Moreover, they carried out a transcriptome analysis, with a focus on phytohormones signaling-related genes, particularly ethylene. The presented study is an interesting piece of research to better understand the Sclerotinia-Brassica interactions and how the pathogen hijacks the plant phytohormones-based defensive response. The manuscript is well-planned and well-written as well. The findings of this study support the conclusion. However, it is only recommended to be published after the following concerns being addressed and corrections being made.

  • Throughout the manuscript the author repeated that “SsERP1 inhibits plant ethylene pathway”, however, I see this statement as confusing and misleading. Which pathway do you mean? It is well-known that all phytohormones have a BIOSYNTHESIS pathway and SIGNALING pathway. As I see here the authors just focused on the signaling pathway (even though it was mentioned very late [Page 9, line 278]). Please be specific and clear. This issue must be revised and corrected throughout the manuscript.
  • As I mentioned above, the authors only focused on the ethylene signaling pathway. What about the ethylene biosynthesis pathway? I highly recommend the authors quantify the ethylene levels, as well as its biosynthetic genes. This will add more value to their work.
  • I have a major concern about the ethylene-jasmonic acid relationship showed in this study. It is well known that ethylene and jasmonic acid are associated together to defend against necrotrophic phytopathogen (ET/JA-mediated pathway). As shown in figure 5C, although the authors showed that the transcript levels of 18 ethylene signaling-related DEGs regulated in Sserp1 mutant-infected napus leaves, however, three JA-related DEGs (TIFY7, COI1, and TIFY10B) were down-regulated. Moreover, the author clearly stated that in their supplementary materials “Ectopic overexpression of SsERP1 did not affect the expression of the jasmonate pathway genes included (NbAOS, NbOPR3, and NbPDF1.2). Kindly, clearly explain this. Moreover, qPCR validation of JA DEGs is required, as you did with ethylene-responsive DEGs (figure 5D). This issue is critical and should be fixed carefully and clearly explained in the revised version of this manuscript.
  • Although the work is well-written, the discussion section is the weakest part of this manuscript. Discussion is very short and insufficient. Discussion must be rewritten with NO mentioning detailed results or figures. Moreover, the ET/JA-mediated pathway must be discussed in light of your findings.
  • A hypothetical model that explains how SsERP1 hijacks/interacts with the plant phytohormones-based defensive response, particularly the ET signaling pathway, will be very helpful and supportive to the conclusion of this study.
  • Finally, Although the language used in the manuscript is easy to follow and understand, however, the manuscript should be carefully and deeply revised for grammar and English use, since some other mistakes were found throughout the whole paper.

Reviewer 2 Report

The study was focused on cloning and characterization of a novel effector SsERP1 (Ethylene Pathway Repressor Protein 1) in Sclerotinia sclerotiorum. SsERP1 is a secretory protein highly expressed at the early stages of S. sclerotiorum infection. The Authors revealed that ectopic over-expression of SsERP1 in plant leaves promoted S. sclerotiorum infection, and the knockout mutants of SsERP1 showed reduced pathogenicity but retained normal mycelial growth and sclerotium formation, suggesting that SsERP1 specifically con-tributes to the pathogenesis of S. sclerotiorum. Transcriptome analysis indicated that SsERP1 promotes S. sclerotiorum infection by inhibiting plant ethylene pathway. Moreover, knock-down SsERP1 by in vitro synthesized double-strand RNAs was able to effectively inhibit S. sclerotiorum infection.

In my opinion the paper is generally interesting. However, I recommend the following improvements:

- I strongly suggest including the electropherograms presenting the RNA bands in agarose gels in the manuscript or in the Supplementary file – it would provide information regarding quality of RNA samples.

- In addition, RIN numbers of RNA samples should be presented in the manuscript (the RNA Integrity Number = RIN).

- The Authors used SYBR Green fluorescent dye during gene expression studies, hence, it is obligatory to perform Melting Curve Analysis, and results of this examination should be added in the manuscript or Supplementary file (e.g., JPG or TIFF file).

- Detailed description of real-time qRT-PCR analyses should be profoundly extended. In addition, there is no information regarding the kinetic method of gene expression quantification (e.g. 2-ΔΔCT method of Livak, Schmittgen 2001?).

- There is the lack of a separate paragraph, describing the statistical analyses used. I suggest using more conservative post-hoc test (e.g. Tukey’s test) instead of Duncan’s test.

- Newer important citations in the research field should be added in References, and older ones should be replaced.

- Discussion of the results is superficial and should be thoroughly revised.

- I suggest a separate paragraph presenting the most relevant conclusions.

- Moderate English changes by the native speaker are required.

Round 2

Reviewer 1 Report

Thanks for addressing most, if not all, of my comments and suggestions.

The only thing that I highly recommend to do is to add the p-values within all bar graphs in this manuscript and to replace the significance asterisks (*) with significance letters  (a, b, c, ... etc). 

Great job and good luck

Author Response

Response to Reviewer  Comments

Point 1: The only thing that I highly recommend to do is to add the p-values within all bar graphs in this manuscript and to replace the significance asterisks (*) with significance letters (a, b, c, ... etc).

Response 1: We have followed your suggestion to replace the significance asterisks with significance letters in all the bar graphs (we didn’t show the p-values for all the bars because of the limit of the graph sizes). We have also revised the figure legends of these bar graphs accordingly. Many thanks.